# Predicting the Risk of Nail Involvement in Psoriasis Patients: Development and Assessment of a Predictive Nomogram

**DOI:** 10.3390/diagnostics13040633

**Published:** 2023-02-08

**Authors:** Yu-Ting Peng, Ren-Tao Yu, Ai-Jun Chen, Zhu-Yuan Wen, Jing Xu, Kun Huang, Ping Wang

**Affiliations:** Department of Dermatology, the First Affiliated Hospital of Chongqing Medical University, Chongqing 400016, China

**Keywords:** psoriasis, nail psoriasis, nail involvement, nomogram, prediction model

## Abstract

Background: Nail involvement has a tremendous impact on psoriasis patients. Early detection and prompt intervention of psoriatic nail damage are necessary. Methods: A total of 4290 patients confirmed to have psoriasis between June 2020 and September 2021 were recruited from the Follow-up Study of Psoriasis database. Among them, 3920 patients were selected and divided into the nail involvement group (*n* = 929) and the non-nail involvement group (*n* = 2991) by inclusion and exclusion criteria. Univariate and multivariable logistic regression analyses were performed to identify the predictors of nail involvement for the nomogram. Calibration plots, the receiver operating characteristic (ROC) curve, and decision curve analysis (DCA) were used to evaluate the discriminative and calibrating ability and clinical utility of the nomogram. Results: Sex, age at onset, duration, smoking, drug allergy history, comorbidity, sub-type of psoriasis, scalp involvement, palmoplantar involvement, genital involvement, and PASI score were selected to establish the nomogram for nail involvement. AUROC (0.745; 95% CI: 0.725–0.765) indicated the satisfactory discriminative ability of the nomogram. The calibration curve showed favorable consistency, and the DCA showed the good clinical utility of the nomogram. Conclusion: A predictive nomogram with good clinical utility was developed to assist clinicians in evaluating the risk of nail involvement in psoriasis patients.

## 1. Introduction

Psoriasis is an immune-related chronic inflammatory disease with worldwide distribution, affecting up to 3% of the general population, which is characterized by epidermal hyper-proliferation resulting in erythematous-squamous skin plaques [1]. Nail involvement is a common but often overlooked feature of the disease, affecting up to 50% of psoriasis patients of various types including psoriatic arthritis (PsA), erythrodermic psoriasis, pustular psoriasis, and psoriasis vulgaris; lifetime incidence of nail psoriasis is 80–90% [2,3]. Nail involvement is highly prevalent in patients with cutaneous psoriasis, and may also present as an isolated finding. Psoriatic nail damage without cutaneous involvement is present in 5–10% of patients [4,5]. The clinical manifestations of psoriatic nail involvement are dependent on which part of the nail anatomy is involved, mainly divided into two types: nail matrix involvement, such as pitting, leuconychia, red spots in the lunula, and nail crumbling; damage to the nail bed, including onycholysis, subungual hyperkeratosis, oil-spot discoloration, and splinter hemorrhages [6,7,8,9,10].

Nail damage is often related to the duration and severity of cutaneous psoriasis [3], and it is frequently observed in association with PsA [11]. Previous studies showed that patients with nail damage usually had higher disease activity than those without nail damage, including severe tender and swollen joint counts, worse disease activity in PsA, and an increased likelihood of having enthesitis and dactylitis [12]. Furthermore, patients with nail damage are more tremendously impacted by the disease burden and have a higher impairment of health-related quality of life (QoL), and psoriatic nail involvement has been well-defined as a factor with a detrimental effect on a psoriasis patient’s QoL by inducing aesthetic problems and functional damage to patients [13,14]. Therefore, identifying potential nail involvement in psoriasis patients based on clinical clues would provide an opportunity for dermatologists to recognize and manage patients at an early time point.

The purpose of this study is to establish an effective and simple prediction model for estimating nail involvement probabilities of diagnosed psoriasis patients by using demographic characteristics and clinical manifestations. We develop a nomogram for the prediction of nail involvement by using the data from China National Clinical Research Center for Skin and Immune Diseases. This prediction model can help with the identification and personalized treatment of high-risk nail-involved psoriasis patients.

## 2. Methods

### 2.1. Study Population

A total of 4260 patients with psoriasis were recruited from 202 assessment centers all over China from June, 2020 to September, 2021, and were observed in the Follow-up Study of Psoriasis. The participants provided information in the baseline assessment through questionnaires and dermatologist assessment, including demographic and clinical characteristics. We finally selected 3920 patients at baseline by the criteria as follows: (1) selecting psoriasis patients with or without nail involvement; (2) at least one of the nails (fingernail or/and toenail) showing typical nail psoriasis signs by naked eye and dermoscopy in nail-involvement patients, including crumbling, onycholysis and oil spot discoloration, pitting, leukonychia, splinter hemorrhages, nail bed hyperkeratosis, red spots in lunula, Beau’s lines/transverse grooves, longitudinal ridges, thickening, and pustule; (3) excluding the patients with onychomycosis, confirmed by microscopic and/or fungal culture examination, or other nail diseases; (4) excluding the patients with missing nail involvement record data. All the patients enrolled were divided into nail involvement (NI) group, of 929 patients, or non-nail involvement (non-NI) group, of 2991 patients, determined by whether nails were involved.

This study used the data from China National Clinical Research Center for Skin and Immune Diseases and has received approval from the Ethics Committee of The First Affiliated Hospital of Chongqing Medical University. As a retrospective study, the informed consent from all enrolled patients was waived under the authorization of the Ethics Committee.

### 2.2. Ascertainment and Assessment

Diagnosis of psoriasis was mainly confirmed using the records obtained from China National Clinical Research Center for Skin and Immune Diseases. Diagnosis of psoriasis was mainly confirmed using the records obtained from China National Clinical Research Center for Skin and Immune Diseases. The International Classification of Diseases (ICD) coding system was used to record the diagnosis of psoriasis. The diagnosis of erythrodermic psoriasis, pustular psoriasis, and psoriasis vulgaris was performed by dermatologists based upon clinical presentation and/or histologic examination. Confirmed diagnosis of PsA was conducted by rheumatologists and dermatologists according to the symptoms of arthritis and Classification Criteria for Psoriatic Arthritis (CASPAR) [15].

Participants provided information on demographic and clinical characteristics in the baseline assessment through questionnaires. The nail clinical manifestations, skin lesions, special parts involvement (scalp, palmoplantar, genital, and arthrosis), Body Surface Area (BSA), Psoriasis Area and Severity Index (PASI) were evaluated by two independent dermatologists. The body mass index (BMI, kg/m^2^) of all the patients was calculated based on their measured weight and height. The QoL and mental health status of psoriasis patients were assessed using the dermatology life quality index (DLQI) [16], and mild impact was defined as DLQI < 6 points, moderate as 6 ≤ DLQI < 10 points, and severe as DLQI ≥ 10 points. The disease severity was assessed using BSA and PASI [17]: mild was defined as BSA < 3% or PASI < 7; moderate as 3% ≤ BSA < 10% or 7 ≤ PASI < 12; and severe as BSA ≥ 10% or PASI ≥ 12. Psoriasis epidemiology screening tool (PEST) is a screening tool for identifying people with PsA [18], and a score of PEST < 3 indicates no or low risk of PsA.

### 2.3. Statistical Analysis

Descriptive statistics were used to assess any difference in patient characteristics, and all the data such as demographic and clinical characteristics were expressed as count (%) or median (IQR). The χ2-test or Fisher’s exact test was used for categorical variables, and the Wilcoxon signed-rank test was used for continuous variables. Univariable analysis was performed to identify the variables associated with nail involvement. A risk model was developed by multivariable logistic regression. We selected variables for inclusion in the nomogram using both statistical significance (*p* < 0.05) and clinical importance criteria. Nail involvement probabilities were estimated using the nomogram. The concordance index (C-index) and receiver operating characteristic (ROC) curve were used to evaluate the discriminative ability of the nomogram. Calibration plots were used to evaluate the calibrating ability, while decision curve analysis (DCA) was used to evaluate the clinical benefits and utility of the nomogram. The statistical significance level was set to be *p* ≤ 0.05, and all *p*-values were two-sided. All statistical analyses were performed using R, version 4.2.0 (https://www.r-project.org/ (accessed on 1 June 2022)).

## 3. Results

### 3.1. Demographic and Clinical Features of NI Group and Non-NI Group

A total of 3920 psoriasis patients were assessed in the present study, and 929(23.70%) participants with nail involvement were in the NI group, while the other 2991 patients (76.30%) without nail damage were in the non-NI group. The baseline characteristics of the participants are shown in Table 1. Overall, there were more males (NI group: 683(73.52%); non-NI group: 1855(62.02%)) than females (NI group: 246(26.48%); non-NI group: 1136(37.98%)). The NI group had a longer duration of psoriasis than the non-NI group (median (IQR): 10.00(14.00) years vs. 6.50(12.00) years). The patients with and without nail damage had a similar age at the onset of psoriasis. The NI group patients were more often active smokers, including current and former smokers, and also had a higher prevalence of psoriasis family history 183(19.70%) vs. 398(13.31%), drug allergy history 79(8.50%) vs. 157(5.25%), light sensitivity 162(17.44%) vs. 424(14.18%), and comorbidities 194(20.88%) vs. 360(12.04%) than those without nail damage. In addition, special parts involvement, including scalp 755(81.27%) vs. 1744(58.31%), palmoplantar 354(38.11%) vs. 426(14.58%), genital 240(25.83%) vs. 282(9.43%), and arthritis 70(7.53%) vs. 94(3.14%) were more common in the patients in the NI group than those in the non-NI group. In addition, the median (IQR) scores of BSA, PASI, PEST, and DLQI in the NI group were significantly higher than those in the non-NI group (Table 1).

### 3.2. Demographic and Clinical Characteristics with DLQI and PASI Score in NI Group

The median (IQR) DLQI and PASI score in the 929 NI patients at baseline was 9.00 [11] and 10.50 [15], respectively. Among the NI group, nail involvement had a large-to-moderate effect on the QoL (DLQI ≥ 6) of 610 (66.37%) patients, and a small effect on QoL (DLQI < 6 = of 309 (33.62%) patients. Meanwhile, 580 (62.44%) patients showed moderate-to-severe psoriasis (PASI ≥ 7) and 325 (71.8%) were mildly affected (PASI < 7). There were some differences in the demographic characteristics and disease characteristics of the DLQI and PASI scores in the NI group (Table 2). The NI patients whose age at onset was < 40 years, with psoriasis family history, scalp involvement, palmoplantar involvement, genital involvement, BSA ≥ 3, PASI ≥ 7, and PEST ≥ 3 had higher DLQI scores. Higher PASI scores were observed in the NI patients with the characteristics of being male, a duration ≥ 10 years, family history, smoking, light sensitivity, special parts involvement (scalp, palmoplantar, and genital), and moderate-to-severe DLQI score. The difference was statistically significant.

### 3.3. Construction of Nail Involvement Psoriasis (NIP) Prediction Model

Among the demographic and clinical characteristics of the 3920 psoriasis patients in the present study, 21 factors were found to be the potential predictors, including sex, age at onset, duration, occupation, education, marital status, BMI, psoriasis family history, smoking, drug allergy history, light sensitivity, comorbidity, sub-type of psoriasis, scalp involvement, palmoplantar involvement, genital involvement, arthrosis involvement, DLQI, BSA, PASI, and PEST. The results of the univariate and multivariate logistic regression analysis are shown in Table 3. The NIP prediction model incorporated the predictors with a significant difference in both the univariate and multivariate logistic regression analyses (*p* < 0.05). After variable selection, ten variables, including sex, age at onset, duration, smoking, drug allergy history, comorbidity, sub-type of psoriasis, special parts involvement (scalp, palmoplantar, and genital), and PASI score were retained in the final model. We constructed a nomogram according to the above-selected features (Figure 1). Based on the unadjusted logistic regression analysis, the seven factors of being male, long psoriasis duration, family history, smoking in the past, smoking now, drug allergy history, and comorbidity were found to increase the risk of nail involvement, with an OR of 1.70(95% CI: 1.44–2.00; *p* < 0.001), 1.60(95% CI: 1.38–1.86; *p* < 0.001), 1.57(95% CI: 1.29–1.90); *p* <  0.001), 2.01 (95% CI: 1.52–2.65; *p* <  0.001), 1.86(95% CI: 1.58–2.18; *p* <  0.001), 1.61(95% CI: 1.21–2.13; *p* <  0.001), and 1.91(95% CI: 1.57–2.31; *p* <  0.001), respectively. The involvement of special parts, such as the scalp, palmoplantar, genital, and arthrosis were also identified as high risk factors for nail involvement, and the corresponding ORs were 3.10(95% CI: 2.58–3.71; *p* <  0.001), 3.63(95% CI: 3.07–4.30; *p* <  0.001), 3.40(95% CI: 2.80–4.12; *p* <  0.001), and 2.51(95% CI: 1.83–3.45; *p* <  0.001), respectively. The prediction model is suitable for psoriasis patients and dermatologists to predict the risk of nail involvement according to the clinical features at present. Each predictive factor in the nomogram was assigned a segment of a different length. The total score was calculated by the nomogram based on the line segment length, which corresponded to a different score; most patients had total risk scores ranging from 140 to 270 in this study. Figure 1 shows an example of using the nomogram to predict the nail involvement probability of a given psoriasis patient. In addition, a simple scoring questionnaire was developed to help dermatologists predict in advance the possibility of psoriatic nail damage (Table 4).

### 3.4. Validation of Nail Involvement Psoriasis Risk Nomogram and Its Clinical Application

The nomogram for predicting nail involvement risk in psoriasis patients was evaluated and validated by the calibration curves, C-index, ROC, and DCA. The calibration curves of the nomogram exhibited high consistencies between the predicted and observed probabilities, as curves moved close to the 45° diagonal line (Figure 2A). The final model was well calibrated with a C-index of 0.75 (95% CI:0.73–0.76). In addition, the ROC curve of the nomogram is presented in Figure 2B and the area under the receiver operating characteristic (AUROC) curve is 0.745 (95% CI: 0.725–0.765).

The DCA indicated that the model proposed has an efficient predictive capability (Figure 2C). The ordinate represents the net benefit, and the abscissa is the threshold probability in the DCA curve. The result of DCA demonstrated that when the screening threshold of nail involvement was given between 0% and 80%, the clinical net benefit would be higher using the proposed nomogram as the screening tool compared to that using the strategies of screening all patients or screening no one. In summary, the nomogram for nail involvement had considerable discriminative and calibrating abilities.

## 4. Discussion

Nail involvement in psoriasis was common but the variability of clinical manifestations and the limitation of therapeutic effects made treatment challenging. Clinicians need to pay attention to nail changes as early as possible. Considering the importance of nail involvement to psoriasis, our study used demographic and clinical characteristics to explore the predictors of nail involvement in psoriasis patients and constructed a prediction model of nail involvement.

In the present study, ten independent variables were identified in univariate and multivariate analyses: sex, age at onset, duration, smoking, drug allergy history, comorbidity, sub-type of psoriasis, scalp involvement, palmoplantar involvement, genital involvement, and PASI. These are readily available in daily clinical practice. The nomogram showed excellent predictive performance for nail involvement and was also validated by the calibration, ROC, and DCA, and thus can be used effectively to screen high-risk patients with nail involvement and to provide a reference for doctors in treatment. According to our nomogram, if a patient achieves a score of 410 or higher, the odds of having nail involvement is >2.1. DCA was used to evaluate whether model-based clinical decisions were effective and the DCA showed the net benefit in the nomogram was better than that in all-patient-positive or all-patient-negative risk scenarios at a threshold probability between 0% and 80%, demonstrating good clinical applicability of this model. According to this result, dermatologists should be highly encouraged to predict the nail condition of psoriasis patients in order to address more appropriate treatment. In addition, these variables are routinely available even in basic hospitals or medically underdeveloped areas, so the prediction model for risk of nail involvement shows suitability for clinical application.

It has been reported that smoking was an independent risk factor for psoriasis, and psoriatic nail was significantly and more frequently observed in smoking psoriasis patients [19,20], which was similar to our results, in part because local angiopathic effects and cigarette rituals (e.g., lighting, gripping cigarettes) lead to koebnerization [20]. In addition, our study showed that comorbidity carried a higher risk of nail involvement and the most common comorbidity was cardiovascular disease, in part because smoking can accelerate the development of comorbidities [21]. In addition, the psoriasis patients with a drug allergy history tended to have a higher risk for nail involvement in our study. Emerging evidence consistently found the association between an increased IL-6 level and drug hypersensitivity reaction [22]; IL-6 may also be associated with NAPSI [23]. This may be related to the fact that both nail psoriasis and drug allergies are associated with the T-cell-mediated immune system, and patients with a drug allergy history were more likely to have a compromised immune system.

A previous study showed the chance of nail involvement in psoriasis was 2.43 times higher in males than in females [24], and Ricardo et al. [25] also reported that 32.1% of participants were female across all nail psoriasis clinical trials. Our study also found that only 26.48% of nail involvement participants were females, although the negative impact of nail psoriasis on QoL among females was greater [26]. The difference in nail involvement by sex may be partly related to less traumatic activities and more aggressive treatment of female patients, so we could not observe obvious nail changes at the recording time. Additionally, Darjani et al. proposed that nail involvement was an important criterion in determining the severity of skin manifestations in psoriatic patients [27]. They found that the PASI scores in nail involvement patients were higher than those in the patients without nail involvement (11.7 ± 5.7 vs. 5.7 ± 4.5). In addition to more severe skin diseases, nail involvement was associated with greater impairment of patients’ QoL together with a longer duration of skin lesions [2,13,28], which were also similar to our results. Importantly, our study indicated that special parts involvement including scalp, palmoplantar, and genital were a highly important risk factor for nail involvement in psoriasis patients but the reasons need to be further explored.

In addition, some studies have revealed that psoriatic nail involvement is closely related to PsA [29,30]. The study by Liu et al. enrolled diagnosed plaque psoriasis and PsA cases in China and demonstrated that nail involvement, erythematous lunula, oil drop, and subungual hyperkeratosis were predictors for PsA [31]. In our study, the ratio of nail involvement in PsA was higher than that in other types of psoriasis, which was similar to previous studies, and the patients with independent PsA had the highest weight in the prediction of nail involvement in all types of psoriasis. However, the proportion of patients with PsA was relatively low in this study. This might be explained by only a few PsA patients seeking medical care in rheumatology and relatively low rates of dermatology visits in the Chinese population of our study.

Treatment of psoriasis with nail involvement may require a longer period before reaching a satisfying therapeutic goal, so nail involvement represents a negative prognostic factor for psoriasis [32]. Therefore, clinicians should pay attention to patients’ nails earlier and develop a more appropriate personalized treatment. Our study established a nail involvement model and provided a clinical basis to help physicians: pay attention to the risk of nail involvement earlier, guide patients’ lifestyles, and prescribe better-individualized treatment regimens.

The strength of this study was the development of an easy-to-apply prediction model to determine the risk of nail involvement in psoriasis patients. This risk prediction nomogram could be used for the initial screening of nail involvement risk. The nomogram had a good effect on clinical practice and could be used for monitoring and personalized treatment of psoriasis patients with a high risk of nail involvement. However, this study still had the following limitations. Firstly, the data lacked more specific nail conditions, so we could not obtain detailed clinical changes and could not know whether the nail matrix, nail bed, or more widespread nails are involved. In addition, the data lacked scoring, which was needed in determining the clinical severity of nail psoriasis, such as the Nail Psoriasis Severity Index (NAPSI), the Nijmegene-Nail Psoriasis Activity Index tool (N-NAIL), and the Psoriasis Nail Severity Score or Nail Area Severity [33]. Furthermore, we only had the DLQI score to assess patients’ mental health status and had no QoL scale to give a specific evaluation of the nail effects of psoriasis, such as the Nail Psoriasis Quality of life 10 (NPQ10) that is a unidimensional scale oriented more towards functional impairment for the patients affected by nail psoriasis [34]. Finally, although this data provided a relatively correct exposure time, the reality was complicated. The patients included in this study were all confirmed as having psoriasis but the time between skin psoriasis or PsA and nail damage was unclear. There may be a small number of patients with nail damage first and without skin or joint involvement, which were not clearly diagnosed with psoriasis and not included in the database. However, these limitations were balanced by the strength of this study and the applicability of the nomogram.

In summary, we developed a predictive model that can give an efficient assessment of the risk of nail involvement in psoriasis patients. Our results showed that sex, age at onset, duration, smoking, drug allergy history, comorbidity, sub-type of psoriasis, scalp involvement, palmoplantar involvement, genital involvement, and PASI were associated with a high risk of nail involvement. Meanwhile, we provided a simple and clear questionnaire to help dermatologists predict in advance the possibility of psoriatic nail damage. In addition, further validation by the statistical test of the random population sample shows that the nomogram could effectively predict the risk of nail involvement through. Furthermore, our study showed the demographic and clinical characteristics with DLQI and PASI scores in nail-involvement patients, which reminds clinicians to pay more attention to nail-involvement patients’ QoL in females, aged < 40 years, with family history, special parts involvement, and severe psoriasis; and to the skin lesions in males, with duration > 10 years, smoking, light sensitivity, and special parts involvement.

## Figures and Tables

**Figure 1 diagnostics-13-00633-f001:**
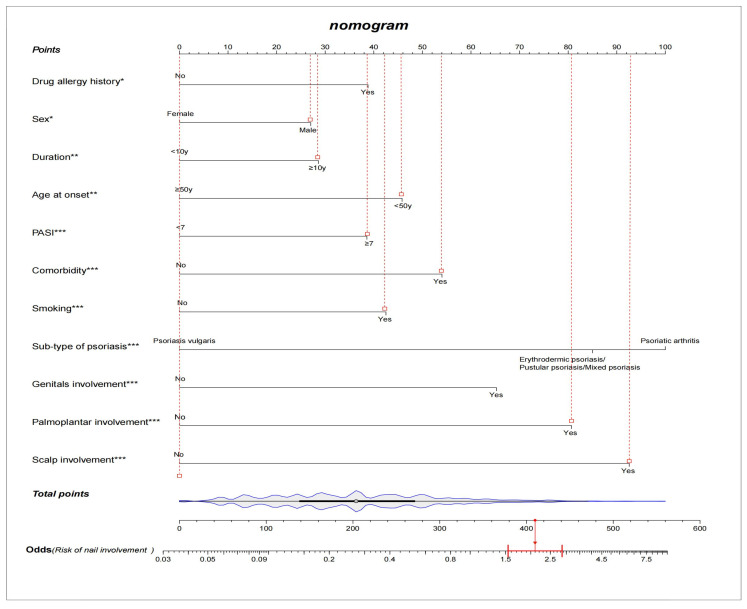
A constructed nomogram for nail involvement prediction of a patient with psoriasis. Density plot of total points shows their distribution. For categorical variables, their distributions are reflected by the length of the line. All variables were ranked according to the *p* value (* *p* value ≤ 0.05; ** *p* value ≤ 0.01; *** *p* value ≤ 0.001). Red lines drawn upward to determine the points received by each variable; the sum of these points is located on the Total Points axis, and a line is drawn downward to the axes to determine the odds of nail involvement. For example, a male patient’s age at onset was 24 years old, and had a psoriasis vulgaris duration for 12 years, with scalp, palmoplantar, and genital involvement. He was smoking, had a drug allergy history, and did not have comorbidity. According to the predictive model, the patient received total points of 410 and the odds of having nail involvement was 2.1.

**Figure 2 diagnostics-13-00633-f002:**
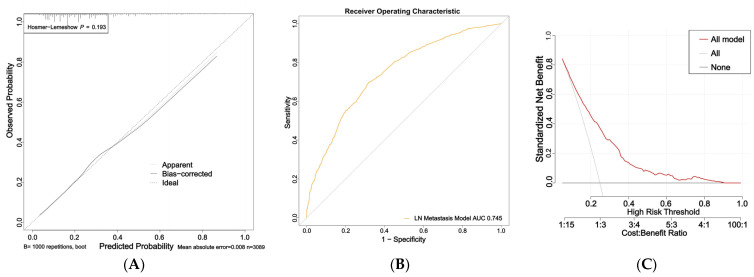
Calibration curves, area under the receiver operating characteristic curve (AUROC), and decision curve of the risk of nail involvement predictive nomogram: (**A**) Calibration curves of the risk of nail involvement predictive nomogram. The *x*-axis represents the predicted risk of nail involvement. The *y*-axis represents the diagnosed nail involvement. The diagonal dotted line represents a perfect prediction of an ideal model. The solid line indicates the performance of the nomogram, and the more consistent the dotted line, the better the prediction ability; (**B**) the AUC of the risk of PsA predictive nomogram. The AUC of the predictive model indicates the probability of accurately predicting whether the patient has nail involvement in a randomly selected case. The model exhibited good predictive power, and the AUC values of the whole sample (orange) are 0.745; (**C**) decision curve of the nomogram based on the risk prediction of patients with nail involvement. The *y*-axis represents the net benefit. The red line indicates the nail-involvement-risk nomogram. The thin solid line represents the assumption that no patients have nail involvement. The gray line represents the assumption that all patients have nail involvement.

**Table 1 diagnostics-13-00633-t001:** Differences between demographic and clinical characteristics of non-nail involvement and nail involvement groups.

Demographic Characteristics	Total (*n* = 3920)	Nail Involvement (*n* = 929)	Non-Nail Involvement (*n* = 2991)	*p* Value
Sex, *n* (%) Female Male	1382(35.26) 2538(64.74)	246(26.48) 683(73.52)	1136(37.98) 1855(62.02)	<0.001
Age at onset, median(IQR)	31.00(21.00)	30.00(20.00)	31.00(21.75)	0.051
Duration(year), median(IQR)	7.00(13.00)	10.00(14.00)	6.50(12.00)	<0.001
BMI, median(IQR)	23.88(4.79)	24.22(4.89)	23.67(4.64)	<0.001
Family history, *n*(%)	581(14.82)	183(19.70)	398(13.31)	<0.001
Smoking, *n*(%) None In the past Now Unknown	2627(67.02) 257(6.56) 1033(26.35) 3(0.08)	519(55.87) 85(9.15) 324(34.88) 1(0.11)	2108(70.48) 172(5.75) 709(23.70) 2(0.07)	<0.001
Drug allergy history, *n*(%)	236(6.02)	79(8.50)	157(5.25)	<0.001
Aggravation seasons, *n*(%) Spring Summer Autumn Winter No obvious seasonality Season change	590(15.09) 304(7.76) 757(19.31) 1852(47.24) 1371(34.97) 270(6.89)	132(14.21) 73(7.86) 155(16.68) 469(50.48) 300(32.29) 68(7.32)	458(15.8) 231(7.72) 602(20.13) 1383(46.24) 1071(35.81) 202(6.75)	0.411 0.893 0.020 0.024 0.050 0.552
Light sensitive, *n* (%)	586(14.95)	162(17.44)	424(14.18)	0.004
Comorbidity, *n* (%) Yes Cardiovascular disease Diabetes Respiratory system disease Hepatopathy disease Gastrointestinal disease Rheumatic immune disease Mental disease Nerve and ENT disease Malignant tumor Renal disease Allergic disease and dermatosis Others NO Unknown	554(14.13) 261(6.66) 131(3.34) 16(0.41) 54(1.38) 25(0.64) 23(0.59) 10(0.26) 6(0.15) 7(0.18) 20(0.51) 28(0.71) 100(2.55) 3076(78.47) 290(7.40)	194(20.88) 91(9.80) 45(4.84) 7(0.75) 14(1.51) 12(1.29) 10(1.08) 3(0.32) 5(0.54) 3(0.32) 7(0.75) 12(1.29) 44(4.74) 678(72.98) 57(7.40)	360(12.04) 170(5.68) 86(2.88) 9(0.30) 40(1.34) 13(0.43) 13(0.43) 7(0.23) 1(0.03) 4(0.13) 13(0.43) 16(0.53) 56(1.87) 2398(80.17) 233(7.79)	<0.001 <0.001 0.004 0.059 0.698 0.004 0.025 0.923 0.003 0.454 0.233 0.017 <0.001
Sub-type of psoriasis, *n* (%) Psoriatic arthritis Erythrodermic psoriasis Pustular psoriasis Psoriasis vulgaris Mixed psoriasis	42(1.07) 82(2.09) 112(2.86) 3575(91.20) 109(2.78)	19(2.05) 28(3.01) 39(4.20) 794(85.47) 49(5.27)	23(0.77) 54(1.81) 73(2.44) 2781(92.98) 60(2.01)	<0.001
Scalp involvement, *n* (%)	2499(63.75)	755(81.27)	1744(58.31)	<0.001
Palmoplantar involvement, *n* (%)	790(20.15)	354(38.11)	426(14.58)	<0.001
Genital involvement, *n* (%)	522(13.32)	240(25.83)	282(9.43)	<0.001
Arthritis involvement, *n* (%)	164(4.18)	70(7.53)	94(3.14)	<0.001
BSA, median(IQR)	10.00(27.00)	15.00(35.00)	10.00(22.00)	<0.001
PASI, median(IQR)	7.00(11.48)	10.50(15.10)	6.00(9.90)	<0.001
PEST, median(IQR)	0.00(1.00)	1.00(2.00)	0.00(0.00)	<0.001
DLQI, median(IQR)	8.00(10.00)	9.00(11.00)	7.00(10.00)	<0.001
Treatment satisfaction for last year, *n*(%) Satisfied Fair Dissatisfied Unknown	1257(32.07) 1865(47.58) 764(19.49) 34(0.87)	250(26.91) 427(45.96) 236(25.40) 16(1.72)	1007(33.67) 1438(48.08) 528(17.65) 18(0.60)	<0.001

**Table 2 diagnostics-13-00633-t002:** Demographic and clinical characteristics with DLQI and PASI score in nail involvement patients.

Characteristics	Patients (*n* = 929)	DLQI	*p* Value	PASI	*p* Value
Sex Male Female	683 246	9.00(3.00,14.00) 10.00(4.00,16.80)	0.050	11.40(5.40,20.10) 7.95(2.80,17.90)	<0.01
Age at onset <40 ≥40	416 513	9.50(4.00,16.00) 9.00(3.00,14.00)	0.047	10.10(4.40,18.80) 11.30(4.65,20.20)	0.148
Duration (year) <10 ≥10	458 471	9.00(3.25,14.00) 9.00(4.00,15.00)	0.142	8.40(3.38,16.20) 13.40(5.90,23.00)	<0.01
BMI <18.5 ≥18.5, <23.9 ≥23.9, <28.0 ≥28.0, <30.0 ≥30.0	23 416 329 81 80	8.00(4.00,15.50) 9.00(3.00,14.50) 9.00(4.00,15.00) 9.00(4.00,15.00) 12.00(5.00,17.50)	0.176	7.80(2.73,18.25) 9.05(3.90,18.90) 11.20(4.85,19.90) 13.90(5.20,21.60) 11.90(6.30,21.60)	0.038
Family history Yes No	183 681	10.00(4.00,16,75) 9.00(3.00,14.00)	0.020	12.30(5.10,21.90) 10.10(4.20,19.00)	0.030
Smoking Yes No	324 604	10.00(4.00,16.00) 9.00(4.00,14.00)	0.189	11.50(6.00,21.60) 9.85(3.90,18.60)	0.002
Drug allergy history Yes No	79 751	8.00(4.00,15.00) 9.00(3.00,15.00)	0.706	10,80(5.40,23.90) 10.40(4.20,19.20)	0.235
Light sensitive Yes No	162 649	10.00(5.00,17.00) 9.00(3.50,15.00)	0.194	14.05(6.95,24.90) 9.90(4.20,18.60)	<0.001
Comorbidity Yes No	194 678	9.00(3.00,15.00) 9.00(3.00,15.00)	0.890	10.65(4.90,21.60) 10.50(4.30,19.10)	0.556
Scalp involvement Yes No	755 171	9.00(4.00,16.00) 6.00(2.00,11.00)	<0.001	11.70(5.70,21.20) 4.60(1.80,12.00)	<0.001
Palmoplantar involvement Yes No	354 563	10.00(5.00,17.00) 8.00(3.00,14.00)	<0.001	15.90(6.75,26.30) 8.50(3.90,26.30)	<0.001
Genital involvement Yes No	240 664	12.00(7.00,17.00) 8.00(3.00,14.00)	<0.001	14.85(6.85,27.75) 9.30(3.90,18.00)	<0.001
Arthrosis involvement Yes No	70 859	8.00(5.00,15.00) 9.00(4.00,15.00)	0.762	11.40(3.50,24.40) 10.40(4.50,19.26)	0.528
DLQI <6 ≥6, <10 ≥10	309 182 428			6.20(2.40,15.00) 9.80(4.76,17.93) 13.80(7.60,25.40)	<0.001
BSA <3 ≥3, <10 ≥10	124 207 572	4.00(1.00,10.00) 7.00(2.00,12.00) 10.00(5.00,17.00)	<0.001	1.80(0.90,2.92) 4.90(3.00,8.00) 16.80(10.25,25.85)	<0.001
PASI <7 ≥7, <12 ≥12	325 176 404	6.00(2,00,11.00) 10.00(5.00,15.50) 11.00(6.00,17.00)	<0.001		
PEST <3 ≥3	749 160	9.00(3.00,14.00) 10.00(5.00,17.50)	0.003	10.40(4.30,19.00) 11.30(4.93,25.30)	0.193
Sub-type of psoriasis Psoriatic arthritis Erythrodermic psoriasis Pustular psoriasis Psoriasis vulgaris Mixed psoriasis	42 82 112 3575 109	9.00(3.00,16.00) 13.00(9.00,18.00) 8.00(3.25,12.00) 8(2.00,12.00) 10(5.00,15.00)	<0.001	8.70(1.58–13.35) 22.80(7.95–31.85) 4.00(1.80–12.60) 6.80(3.00–14.00) 9.50(3.90–24.05)	<0.001

**Table 3 diagnostics-13-00633-t003:** The multivariate and univariate logistic regression analysis of nail involvement.

Intercept and Variable	Univariate Analysis			Multivariate Analysis		*p*-Value
	β	OR (95% CI)	*p*-Value	β	OR (95% CI)	
Intercept						
Sex Female Male	0.53	Ref 1.70(1.44–2.00)	<0 .001	0.27	Ref 1.31(1.05–1.64)	0.017
Age at onset <50 ≥50	0.28	1.33(1.07–1.64) Ref	0.009	0.47	1.60(1.21–2.21) Ref	<0 .001
Duration (months) <120 ≥120	0.47	Ref 1.60(1.38–1.86)	< 0.001	0.29	Ref 1.34(1.11–1.62)	0.002
BMI(China) <18.5 ≥18.5, <23.9 ≥23.9, <27.0 ≥27.0, <30.0 ≥30.0	0.58 0.63 0.98 0.63	Ref 1.78(1.13–2.80) 1.87(1.18–2.98) 2.68(1.65–4.33) 1.88(1.13–3.13)	< 0.001 0.013 0.008 <0 .001 0.015			
Family history	0.45	1.57(1.29–1.90)	<0 .001			
Smoking No In the past Now	0.70 0.62	Ref 2.01(1.52–2.65) 1.86(1.58–2.18)	<0 .001 <0 .001 <0 .001	0.41 0.42	Ref 1.51(1.06–2.15) 1.52(1.22–1.88)	0.022 < 0.001
Drug allergy history	0.47	1.61(1.21–2.13)	0.001	0.41	1.50(1.08–2.10)	0.025
Light sensitive	0.20	1.23(1.00–1.50)	0.048			
Comorbidity	0.64	1.91(1.57–2.31)	<0 .001	0.55	1.73(1.37–2.19)	<0 .001
Sub-type of psoriasis Psoriasis vulgaris Psoriatic arthritis Erythrodermic psoriasis Pustular psoriasis Mixed psoriasis	1.06 0.60 0.63 1.05	Ref 2.89(1.57–5.34) 1.82(1.14–2.89) 1.87(1.26–2.78) 2.86(1.95–4.21)	< 0.001 0.012 0.002 <0 .001	1.14 0.03 0.87 0.83	Ref 3.12(1.41–6.94) 1.04(0.56–1.92) 2.39(1.45–3.94) 2.30(1.43–3.70)	0.005 0.912 < 0.001 <0 .001
Scalp involvement	1.13	3.10(2.58–3.71)	<0 .001	0.91	2.49(1.99–3.11)	<0 .001
Palmoplantar involvement	1.29	3.63(3.07–4.30)	< 0.001	0.80	2.23(1.80–2.76)	<0 .001
Genital involvement	1.22	3.40(2.80–4.12)	<0 .001	0.65	1.91(1.50–2.43)	<0 .001
Arthrosis involvement	0.92	2.51(1.83–3.45)	<0 .001			
DLQI <6 ≥6, <10 ≥10	0.04 0.46	Ref 1.04(0.85–1.28) 1.58(1.34–1.87)	<0 .001 0.688 < 0.001			
BSA <3 ≥3, <10 ≥10	0.35 0.70	Ref 1.42(1.11–1.81) 2.01(1.62–2.49)	<0 .001 0.005 <0 .001			
PASI <7 ≥7	0.75	Ref 2.11(1.81–2.47)	<0 .001	0.38	Ref 1.47(1.22–1.77)	<0 .001
PEST <3 ≥3	1.34	Ref 3.84(3.03–4.85)	<0 .001			

**Table 4 diagnostics-13-00633-t004:** A simple scoring questionnaire to predict psoriatic nail damage based on nomogram. The total score is 86, higher scores indicate a higher risk of nail damage.

Risk Factor	Addition to Risk Score
Sex (male)	3
Duration of psoriasis (year) ≥ 10	3
Drug allergy history	4
PASI ≥ 7	4
Smoking	4
Age at onset (year) < 50	5
Comorbidity	5
Genital involvement	7
Palmoplantar involvement	8
Scalp involvement	9
Sub-type of psoriasis Erythrodermic psoriasis Pustular psoriasis Mixed psoriasis Psoriatic arthritis	8 8 8 10
Total score	

## Data Availability

The raw data supporting the conclusions of this article will be made available by Yuting Peng, without undue reservation.

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
