# Peer review of "Predicting the Risk of Nail Involvement in Psoriasis Patients: Development and Assessment of a Predictive Nomogram"

_diagnostics, 2023, doi:10.3390/diagnostics13040633_

Round 1
Reviewer 1 Report
The study is very interesting. However, I think that the authors should evaluate each group of comorbidities (such as cardiovascular, pulmonary etc) separately.
Author Response
Thank you very much for your suggestion. In the revised version, we evaluated the comorbities of psoriasis, separately. Details were showed in table one and page four of the revised version.
Reviewer 2 Report
In this study, the authors want to establish an effective and simple prediction model for estimating nail involvement in psoriasis patients. Their model derived by a statistical analysis, but I suggest to insert a simple and clear questionnaire to help dermatologist in in a predictive diagnosis
- Line 29, 31, 32, 35, 36, 40 : put a space between the word and the round parenthesis
of the number
· In the present study to develop a smart predictive model that can give an efficient assessment of the risk of nail involvement in psoriasis patients, it would be better to organize and add in this paper a questionnaire to help dermatologists prediction in advance the possibility of psoriatic nail damage according to all the parameters you have identified and described
Author Response
Thank you very much for your suggestion. We have checked and revised the article and here are our replies.
For the evaluation form:
- In the revised version, complements and modifications are made for introduction.
- We checked and revised references, and all references given are relevant to the research.
- In the revised version, some details about this research were added in the methods part.
- According to your precious suggestion, in the revised version, we added a simple scoring questionnaire for predictive psoriatic nail damage.
For the Comments and Suggestions:
- Thank you for your careful advice, in the revised version, we have already put a space between the word and the round parenthesis of the number at Line 29, 31, 32, 35, 36, 40.
- As mentioned above, we have added a questionnaire to help dermatologist in a predictive psoriatic nail damage.
Round 2
Reviewer 2 Report
Many Thank to correct the work according my comments, so the paper can be accepted in this form.